# Effects of 4:3 Intermittent Fasting on Eating Behaviors and Appetite Hormones: A Secondary Analysis of a 12-Month Behavioral Weight Loss Intervention

**DOI:** 10.3390/nu17142385

**Published:** 2025-07-21

**Authors:** Matthew J. Breit, Ann E. Caldwell, Danielle M. Ostendorf, Zhaoxing Pan, Seth A. Creasy, Bryan Swanson, Kevin Clark, Emily B. Hill, Paul S. MacLean, Daniel H. Bessesen, Edward L. Melanson, Victoria A. Catenacci

**Affiliations:** 1Division of Endocrinology, Metabolism, and Diabetes, Department of Medicine, University of Colorado Anschutz Medical Campus, 12348 E. Montview Boulevard, Aurora, CO 80045, USA; ann.caldwell@cuanschutz.edu (A.E.C.); danielle.ostendorf@cuanschutz.edu (D.M.O.); seth.creasy@cuanschutz.edu (S.A.C.); paul.maclean@cuanschutz.edu (P.S.M.); daniel.bessesen@cuanschutz.edu (D.H.B.); ed.melanson@cuanschutz.edu (E.L.M.); vicki.catenacci@cuanschutz.edu (V.A.C.); 2Anschutz Health and Wellness Center, University of Colorado Anschutz Medical Campus, 12348 E. Montview Boulevard, Aurora, CO 80045, USA; kevin.2.clark@cuanschutz.edu (K.C.);; 3Department of Kinesiology, Recreation and Sports Studies, University of Knoxville, Knoxville, TN 37996, USA; 4Department of Biostatistics & Informatics, University of Colorado Anschutz Medical Campus, Aurora, CO 80045, USA; zhaoxing.pan@cuanschutz.edu; 5School of Medicine, University of Colorado Anschutz Medical Campus, Aurora, CO 80045, USA; bryan.swanson@cuanschutz.edu; 6Department of Pediatrics, Section of Nutrition, University of Colorado Anschutz Medical Campus, Aurora, CO 80045, USA; 7Division of Geriatric Medicine, Department of Medicine, University of Colorado Anschutz Medical Campus, Aurora, CO 80045, USA

**Keywords:** appetite regulation, binge eating, uncontrolled eating, behavioral intervention, caloric restriction

## Abstract

Background/Objectives: Daily caloric restriction (DCR) is a common dietary weight loss strategy, but leads to metabolic and behavioral adaptations, including maladaptive eating behaviors and dysregulated appetite. Intermittent fasting (IMF) may mitigate these effects by offering diet flexibility during energy restriction. This secondary analysis compared changes in eating behaviors and appetite-related hormones between 4:3 intermittent fasting (4:3 IMF) and DCR and examined their association with weight loss over 12 months. Methods: Adults with overweight or obesity were randomized to 4:3 IMF or DCR for 12 months. Both randomized groups received a matched targeted weekly dietary energy deficit (34%), comprehensive group-based behavioral support, and a prescription to increase moderate-intensity aerobic activity to 300 min/week. Eating behaviors were assessed using validated questionnaires at baseline and months 3, 6, and 12. Fasting levels of leptin, ghrelin, peptide YY, brain-derived neurotrophic factor, and adiponectin were measured at baseline and months 6 and 12. Linear mixed models and Pearson correlations were used to evaluate outcomes. Results: Included in this analysis were 165 adults (mean ± SD; age 42 ± 9 years, BMI 34.2 ± 4.3 kg/m^2^, 74% female) randomized to 4:3 IMF (n = 84) or DCR (n = 81). At 12 months, binge eating and uncontrolled eating scores decreased in 4:3 IMF but increased in DCR (*p* < 0.01 for between-group differences). Among 4:3 IMF, greater weight loss was associated with decreased uncontrolled eating (r = −0.27, *p* = 0.03), emotional eating (r = −0.37, *p* < 0.01), and increased cognitive restraint (r = 0.35, *p* < 0.01) at 12 months. There were no between-group differences in changes in fasting appetite-related hormones at any time point. Conclusions: Compared to DCR, 4:3 IMF exhibited improved binge eating and uncontrolled eating behaviors at 12 months. This may, in part, explain the greater weight loss achieved by 4:3 IMF versus DCR. Future studies should examine mechanisms underlying eating behavior changes with 4:3 IMF and their long-term sustainability.

## 1. Introduction

Overweight and obesity represent a major global health crisis, with projections estimating that 3.8 billion adults will be affected by 2050 unless effective prevention and treatment strategies are implemented [1]. Daily caloric restriction (DCR) is the standard dietary approach in behavioral weight loss interventions for treating overweight and obesity [2]. However, DCR triggers compensatory physiological and metabolic adaptations, including the dysregulation of appetite-related hormones, maladaptive eating behaviors, and a heightened hunger response, which together, promote increased energy intake (EI) and weight regain [3,4,5]. Intermittent fasting (IMF) is an alternative dietary weight loss approach that involves cycling between complete (100%) or near-complete (>75%) energy restriction on “fast” days and *ad libitum* EI on non-fasting days. Various IMF paradigms have been proposed, including two non-consecutive fast days per week (5:2 IMF), three non-consecutive fast days per week (4:3 IMF), and fasting every other day (alternate day fasting; ADF). To date, five long-term (≥12 months) randomized trials have compared IMF paradigms to DCR (four trials using 5:2 IMF and one trial using ADF) with matched targeted weekly energy deficits, and all found no significant difference in weight loss between groups [6,7,8,9,10]. Compared to 5:2 IMF, 4:3 IMF is likely to produce a greater weekly energy deficit due to an additional fast day per week. Further, 4:3 IMF allows self-selection of fast days, unlike ADF, potentially offering a more flexible approach and enhancing adherence. We recently reported 4:3 IMF produced significantly greater dietary adherence (assessed with the doubly labeled water [DLW] intake-balance method) and significantly greater weight loss at 12 months as compared to DCR, matched for a targeted weekly energy deficit [11]. However, the mechanisms behind the superior dietary adherence and weight loss in 4:3 IMF compared to DCR remain unclear.

Eating behaviors are defined as the patterns and actions related to food consumption, shaped by a dynamic interplay of physiological, psychological, social, and genetic factors influencing food preference, meal timing, and portion sizes [12,13]. Maladaptive eating behaviors, including emotional eating (i.e., a tendency to eat in response to negative emotions), uncontrolled eating (i.e., a tendency to lose control over food intake), and binge eating behaviors (i.e., consuming large amounts of food in a discrete period, followed by a sense of loss of control) are strongly linked to obesity and weight gain [14,15,16] and can be exacerbated by caloric restriction during weight loss [17,18]. While some observational and cross-sectional studies suggest that IMF is associated with increased disordered eating [19,20,21,22], findings from prospective, interventional studies indicate IMF may improve [23,24,25] or have no effect [9,26,27] on eating behaviors compared to DCR. These conflicting outcomes may reflect differences in intervention duration, behavioral support, participant characteristics, or study design. Moreover, most prior studies have been short-term (e.g., ≤6 months) and there remains a lack of evidence regarding the long-term effects (i.e., ≥12 months) of IMF on eating behaviors.

IMF may also improve appetite regulation by modifying orexigenic and anorexigenic appetite-related hormones. Orexigenic hormones (e.g., ghrelin) stimulate hunger and food-seeking behavior [28], while anorexigenic hormones (e.g., leptin, adiponectin, peptide-YY [PYY], and brain-derived neurotrophic factor [BDNF]) suppress hunger and promote satiety. Recent systematic reviews and meta-analyses indicate that IMF has a beneficial impact on appetite-related hormones by increasing adiponectin and decreasing leptin, ghrelin, and PYY [29,30,31]. While IMF has been shown to elevate levels of BDNF, which is associated with enhanced cognitive control of eating and reduced food reward sensitivity [32,33], some studies have reported conflicting results regarding the effects of IMF on BDNF levels [34]. Given the mixed evidence and lack of long-term trials comparing IMF and DCR on appetite, more research is needed to elucidate the potential mediators of energy balance via appetite regulation and eating behaviors in IMF vs. DCR.

In this secondary analysis, we explored the associations between eating behaviors, appetite hormones, and weight loss in a 12-month randomized trial designed to compare weight loss generated by 4:3 IMF and DCR. By investigating potential behavioral and physiological mechanisms underlying weight loss outcomes, our findings may inform more effective and sustainable obesity treatment strategies globally.

## 2. Material and Methods

### 2.1. Trial Registration

This study involved a secondary analysis of the Daily Caloric Restriction versus Intermittent Fasting Trial (DRIFT), a 12-month randomized clinical trial conducted from 2018 to 2022 at the University of Colorado Anschutz Medical Campus (CU AMC). The study was approved by the Colorado Multiple Institutional Review Board (COMIRB) and registered at ClinicalTrials.gov (NCT03411356), with 12-month primary outcomes published previously [11]. All participants provided written informed consent and were financially compensated.

### 2.2. Study Participants

Adults (aged 18–60 years) with a body mass index (BMI) of 27–46 kg/m^2^ were eligible to participate in the patent trial. Full inclusion and exclusion criteria and methods have been published previously [35].

In brief, participants were excluded if they had a history of cardiovascular disease, diabetes, stage 4–5 chronic kidney disease, major depression or other significant psychiatric disorder, bariatric surgery, use of anti-obesity medications or other medications known to significantly affect body weight, or >5 kg weight change in the previous 3 months. Women who were pregnant, lactating, or planning to become pregnant were also excluded. Participants were also excluded if they had a history of clinically diagnosed eating disorders, had scores ≥ 20 on the Eating Attitudes Test-26 [36], or screened positive for recurrent binge-eating episodes (≥1/week for at least 3 months) or related behaviors (≥3 of 5 associated features) on the Questionnaire on Eating and Weight Patterns (QEWP-5) [37].

### 2.3. Randomization and Interventions

Participants were randomized (1:1) to either 4:3 IMF or DCR for 12 months. The prescribed weekly dietary energy deficit from baseline weight maintenance energy requirements was the same for both groups (34.3%). Maintenance energy requirements were calculated by multiplying resting energy expenditure, measured using indirect calorimetry, by a standardized activity factor of 1.5 [38]. Participants in the 4:3 IMF group were instructed to restrict EI by 80% from estimated baseline weight maintenance energy needs on three non-consecutive days/week with ad libitum intake the other four days. Participants in the DCR group were instructed to reduce daily EI by 34.3% to match the weekly target dietary energy deficit of 4:3 IMF. Participants in both groups were prescribed a targeted dietary macronutrient intake of 55% carbohydrates, 15% protein, and 30% fat. DCR participants were instructed to count calories and log food intake daily to support adherence to the prescribed daily energy restriction. The 4:3 IMF participants also received instruction in calorie counting and food logging, but were instructed to log intake only on fast days. Both groups were advised to progressively increase moderate-intensity aerobic activity to 300 min/week during the first 6 months and maintain this level of activity throughout the study.

### 2.4. Behavioral Support

Both groups received a high-intensity, group-based behavioral weight loss program with equivalent contact and support over the 12-month intervention that has been described previously [35]. The intervention was designed in accordance with current clinical guidelines for behavioral obesity treatment [2]. Randomized groups attended separate sessions facilitated by a registered dietitian nutritionist (RDN) with expertise in facilitation of group-based behavioral weight loss interventions. The curriculum for both groups was adapted from the Colorado Weigh program, which integrates a skills-based methodology and cognitive behavioral strategies to promote sustained lifestyle modification [39,40]. Additional content was incorporated into the curriculum for each group to focus on DCR or 4:3 IMF. Group sessions occurred weekly for the first 3 months and bi-weekly from months 4–12. Each session was approximately 60 min and consisted of large group discussions, small breakout groups, and visual and written demonstrations and exercises. Participants in the 4:3 IMF group were taught strategies to manage hunger specifically on fast days including distraction techniques, flexible meal timing (e.g., concentrating intake at dinner), and portion size control. In contrast, participants in the DCR group were instructed in daily strategies to support continuous calorie restriction, with an emphasis on portion size awareness and daily calorie counting and food logging.

### 2.5. Assessments

#### 2.5.1. Eating Behaviors

Eating behaviors were assessed using the following validated questionnaires completed by participants at baseline and months 3, 6, and 12.

##### Three-Factor Eating Questionnaire

The Three-Factor Eating Questionnaire—Revised 18-item (TFEQ-R18) was used to assess three key dimensions of eating behavior: uncontrolled eating (loss of control over food intake with subjective feelings of hunger; 9 items), emotional eating (lack of ability to resist emotional cues; 3 items), and cognitive restraint (conscious restriction of food intake for weight management; 6 items) [41]. Each item is rated on a 4-point Likert scale (1 = strongly disagree, 4 = strongly agree), with higher scores indicating greater tendencies toward the corresponding eating behavior. Subscale scores were combined to create a scale ranging from 0 to 100. The summed raw scores were then normalized using the following formula: Normalized Score = 100 × (Raw – Min(Raw))/[Max(Raw) − Min(Raw)], where “Raw” represents the summed raw score, and “Min(Raw)” and “Max(Raw)” are the minimum and maximum possible values of the summed raw scores, respectively. Higher scores indicate greater levels of uncontrolled eating, emotional eating, or cognitive restraint, respectively.

##### Binge Eating Scale

The Binge Eating Scale (BES) is a 16-item self-report questionnaire that evaluates the frequency, severity, and emotional consequences of binge eating episodes (vs. binge eating disorder diagnosis), with a particular focus on feelings of loss of control, distress, and physical discomfort during overeating [42]. Each item contains 3 or 4 weighted response options, reflecting a range of severity for each measured characteristic. Item scores range from 0 to 3 or 0 to 4 and were summed for a possible total of 46, with higher scores indicating more severe binge eating behaviors.

##### Reward-Based Eating Drive Scale

The Reward-Based Eating Drive Scale, Revised 13-item (RED-13) is an expanded version of the RED-9, initially created to assess three dimensions of reward-related eating (lack of satiety, preoccupation with food, and lack of control over eating) [43]. Each item is rated on a 5-point Likert scale (0 = strongly disagree, 4 = strongly agree). Scores were summed for a possible total of 52, with higher scores indicating a greater tendency toward reward-related eating.

#### 2.5.2. Appetite Hormones

Blood samples were obtained at baseline and at months 6 and 12 in the morning following a 12 h overnight fast. In the 4:3 IMF group, blood samples were collected after a “non-fast” day to provide within-group consistency. Blood samples were centrifuged for 10 min at 1000× *g* and 4 °C and then stored at −80 °C until analysis. Leptin, ghrelin, PYY, and adiponectin were measured using commercially available radioimmunoassay kits (Millipore, St. Charles, MO, USA) according to the manufacturer’s specifications. Intra- and inter-assay coefficients of variation were 5.9 and 5.8% for leptin, 4.5 and 13.5% for total ghrelin, 5.3 and 8.9% for PYY, and 5.1 and 7.3% for adiponectin. The ratios between adiponectin and leptin concentrations were used to determine the adiponectin/leptin ratio index. This ratio is considered a reliable biomarker of adipose tissue dysfunction and systemic inflammation, with lower values associated with increased cardiometabolic risk and obesity-related complications [44]. Serum BDNF was measured using an ELISA kit (Aviscera Bioscience, Santa Clara, CA, USA). Intra- and inter-assay CVs were 5.7% and 7.8%, respectively. All samples, standards, and controls were plated in duplicate, which were used to create a standard curve. Quality controls fell within expected limits for all assays.

### 2.6. Statistical Analysis

Data was analyzed using SAS version 9.4 (Cary, NC, USA). Linear mixed models with an unstructured covariance were used to examine changes in continuous outcomes within and between randomized groups over 12 months using an intent-to-treat analysis. Associations between the change in body weight, appetite hormones, and eating behavior scores over 12 months were determined using Pearson correlation coefficients. Between-group differences at each time point were assessed using linear mixed-effects models with pre-specified contrasts, with change from baseline to 12 months designated as the primary outcome. This secondary analysis did not include a formal power calculation, as it was exploratory in nature. However, the parent randomized controlled trial was powered at 90% (α = 0.05) to detect a between-group difference of 3 kg in body weight change at 12 months from baseline. All five eating behavior and appetite-related hormone outcomes were considered secondary and exploratory; therefore, no formal adjustment for multiple comparisons was applied to the a priori hypothesis tests.

## 3. Results

### 3.1. Baseline Characteristics

A total of 165 participants were randomized into either 4:3 IMF (n = 84) or DCR (n = 81). At 12 months, the overall attrition rate was 24% (19% in 4:3 IMF and 30% in DCR, see CONSORT diagram in parent trial [11]). Baseline characteristics of randomized participants are shown in Table 1. There were no significant differences in mean baseline eating behavior scores or appetite hormones between groups. The number of participants with available data for eating behavior questionnaires and appetite-related hormone assessments at each time point is shown in Table 2.

### 3.2. Change in Eating Behavior Scores over 12 Months

Figure 1 displays changes in eating behavior scores across 12 months by intervention arm. There were significant group x time interactions for BES scores (Figure 1A, *p* < 0.01) and uncontrolled eating subscale scores (Figure 1B, *p* < 0.01). Post hoc comparisons showed that participants in the 4:3 IMF group reported significantly greater reductions in BES and uncontrolled eating scores from baseline to month 12 (*p* < 0.01) and months 6 to 12 (*p* < 0.01) compared to DCR. There were no significant group × time interactions for emotional eating, cognitive restraint, or reward-based eating scores. A significant main effect of time was observed for emotional eating (Figure 1C, *p* < 0.01), cognitive restraint (Figure 1D, *p* < 0.01), and reward-based eating (Figure 1E, *p* < 0.01), indicating overall change irrespective of group. Within-group comparisons showed that participants in the 4:3 IMF group experienced significant reductions (i.e., improvement) in the emotional eating scores (mean [95% CI]; −5.9 [−11.1 to −0.8]; *p* = 0.03) and reward-based eating (−2.7 [−4.5 to −1.0]; *p* < 0.01) at 12 months compared to baseline; these changes were not observed in the DCR group (Appendix A). At month 3, the DCR group showed a greater increase in cognitive restraint scores compared to the 4:3 IMF group (Appendix A, mean [95% CI]; 18.3 [14.5 to 22.2] vs. 12.8 [9.1 to 16.6]; *p* < 0.05).

### 3.3. Correlation Between Eating Behavior Scores and Weight Loss

Correlations between changes in body weight, TFEQ-R18, BES, and RED-13 eating behavior scores over 12 months are shown in Figure 2. Among 4:3 IMF participants, weight loss was associated with decreases in TFEQ-R18 subscale scores of uncontrolled eating (Figure 2A, *p* = 0.03), emotional eating (Figure 2C, *p* < 0.01), and increases in cognitive restraint (Figure 2E, *p* < 0.01). Among DCR, weight loss was associated only with increases in cognitive restraint (Figure 2F, *p* = 0.01). Changes in the BES and RED-13 scores were not significantly associated with weight loss for either group.

### 3.4. Change in Appetite-Related Hormones over 12 Months

Mean fasting levels of leptin, ghrelin, PYY, BDNF, adiponectin, and the adiponectin/leptin ratio at each time point, as well as change from baseline are shown in Table 3. There were no significant group x time interactions for changes in appetite-related hormones between 4:3 IMF and DCR over 12 months. There were significant time effects for leptin, ghrelin, and the adiponectin/leptin ratio, with no differences between groups. There were significant time and group effects for adiponectin, indicating that adiponectin increased more in 4:3 IMF than DCR over 12 months.

## 4. Discussion

We recently found that 4:3 IMF resulted in greater dietary adherence (objectively measured % caloric restriction; assessed with the DLW intake-balance method) and superior weight loss as compared to DCR over 12 months in 165 adults with overweight or obesity [11]. In this secondary analysis of data from this randomized clinical trial, we found that 4:3 IMF was associated with greater improvements in binge eating behaviors and uncontrolled eating compared to DCR at 12 months. Importantly, improvements in uncontrolled eating and emotional eating behaviors were associated with greater weight loss in the 4:3 IMF group, while no such associations were observed in the DCR group. There were no between-group differences in changes in fasting appetite hormones over 12 months. These findings suggest that improved eating behaviors may have played a role in the superior weight loss outcomes observed with 4:3 IMF and provide novel insight into how different dietary approaches may influence the psycho-behavioral mechanisms underlying dietary adherence and success in weight management.

Successful long-term weight loss and maintenance requires persistent adherence to a dietary pattern that induces a negative energy balance [45]. Our results show that differences in eating behaviors occurred between months 6 and 12, which likely reflect the long-term psychological and behavioral demands of each dietary approach. In the 4:3 IMF group, improvements in the uncontrolled eating and emotional eating scores were significantly correlated with weight loss over 12 months, suggesting that reductions in these maladaptive eating behaviors may have facilitated greater dietary adherence and, consequently, greater weight loss. These associations were not observed in the DCR group, underscoring the potential importance of behavioral flexibility in supporting sustained weight control. The periodic and self-selected nature of the fasting schedule in 4:3 IMF may have reduced perceived deprivation, psychological stress, and food-related preoccupations. In contrast, DCR requires daily calorie tracking and constant monitoring, which imposes a sustained cognitive load and heightened focus on food. The inherent flexibility of 4:3 IMF may have allowed participants to better regulate hunger and satiety cues, reducing the psychological strain often associated with rigid dietary restraints like those found with DCR [46]. The divergence in eating behavior scores may reflect the greater psychological demands of DCR. These include perceived rigidity and deprivation, which are recognized as risk factors for disordered eating behaviors [17,47].

Previous reports have suggested the potential for IMF to promote binge eating and maladaptive eating behaviors [48,49,50]. However, our results over a 12-month intervention in individuals without a history of eating disorders suggest that 4:3 IMF does not exacerbate, but rather, reduces binge eating and uncontrolled eating behaviors compared to DCR. These findings contrast with short-term studies that evaluated ADF or 5:2 IMF, which showed similar improvements in binge eating and uncontrolled eating behaviors in both IMF and DCR approaches over 3 months [26,51]. The divergence observed in our long-term study highlights the importance of duration and dietary flexibility in shaping long-term behavioral outcomes. Notably, participants in our study received structured dietary counseling, which may have supported the observed improvements and mitigated potential risks. To the best of our knowledge, this is the first study to examine the longer-term (≥1 year) effect of a 4:3 IMF paradigm compared to DCR on binge eating behaviors and uncontrolled eating.

Only one other clinical trial to date has evaluated the effects of IMF on eating behaviors. Bhutani et al. investigated an ADF intervention, with or without exercise, and reported reductions in emotional and uncontrolled eating over 12 weeks [24]. Similar to our study, their intervention included regular dietary counseling and behavioral support, which they identified as a key contributor to the observed improvements in eating behaviors. Behavioral interventions are known to enhance adherence and reduce maladaptive eating behaviors [52], suggesting that structured support in both trials may have been critical in promoting healthier eating behaviors. In contrast, the continuous calorie tracking required in DCR may lead to cognitive fatigue, potentially diminishing the effectiveness of behavioral counseling compared to the more flexible 4:3 IMF approach.

Cognitive restraint refers to the tendency to intentionally limit EI to maintain or induce weight loss. Both 4:3 IMF and DCR demonstrated increases in cognitive restraint over the 12 months, indicating a deliberate effort to control their eating. Weight loss in both 4:3 IMF and DCR was positively associated with greater cognitive restraint, consistent with studies that have linked restraint to successful weight loss maintenance [52,53,54]. The higher intensity of behavioral counseling during the initial intervention phase, characterized by weekly group sessions through month 3, may have reinforced cognitive restraint in both groups, leading to greater dietary adherence and successful weight loss. Following the transition to biweekly sessions from months 4–12, this early behavioral momentum may have attenuated, potentially contributing to subsequent declines in dietary adherence and restraint. Prior studies have also found that IMF can lead to significant increases in cognitive restraint within the first 3 months, which then tend to stabilize over time, as was observed in our study [23,24]. Interestingly, the DCR group exhibited a more pronounced increase in cognitive restraint at 3 months compared to 4:3 IMF, which may indicate a greater reliance on rigid control strategies (i.e., an “all-or-nothing” approach to dieting) as opposed to flexible restraint. This type of inflexible restraint has been associated with greater psychological strain and a higher risk of disordered eating patterns, which can undermine long-term success if lapses occur. Similar findings were reported in a 3-month study comparing DCR and 5:2 IMF, where DCR exhibited greater early rises in cognitive restraint scores [51]. This suggests that the more flexible nature of 4:3 IMF may better support sustained behavior change by reducing the psychological burden that often accompanies strict dietary control [47,55]. Overall, these findings underscore the importance of adopting a flexible dietary approach as a more sustainable strategy for achieving long-term weight loss.

The 4:3 IMF group exhibited sustained reductions in emotional eating scores over 12 months, whereas the DCR group showed an initial decline followed by a return to baseline levels by month 12. Notably, reductions in emotional eating were significantly associated with weight loss in the 4:3 IMF group only. These findings are consistent with previous short-term studies showing reductions in emotional eating following an ADF protocol [24] as well as both 5:2 IMF and DCR interventions over 3 months [51]. To our knowledge, this is the first long-term study to demonstrate that 4:3 IMF produces sustained reductions in emotional eating over 12 months, and that these reductions are specifically associated with weight loss.

Our results indicate both 4:3 IMF and DCR decreased reward-based eating over 12 months; however, weight loss was not associated with reward-based eating scores in either group. The RED-13 assesses an individual’s drive to eat as influenced by reward sensitivity (i.e., the degree of eating for pleasure). Functional magnetic resonance imaging (fMRI) studies have shown that acute fasting biases brain reward systems toward reward-seeking behaviors [56]. A recent fMRI study in adults with obesity used an intermittent energy restriction protocol involving alternating unrestricted days with progressively lower EI, followed by a phase of very low-calorie intake (500–600 kcal/d) every other day. This intervention led to a reduced drive to eat, lower neural response to food cues, and improved self-control [57]. Although this approach differs from the 4:3 IMF regimen used in our study, the findings support the idea that sustained energy restriction may reduce reward-based eating. Taken together, these results indicate that both 4:3 IMF and DCR may dampen neural and behavioral drivers of reward-based eating. The findings from our analysis using questionnaire-based assessments of reward-based eating suggest that changes were similar between 4:3 IMF and DCR over 12 months, indicating that both dietary approaches may help reduce the drive to eat for reward. Future studies should evaluate the impact of 4:3 IMF versus DCR on neurobehavioral reward processes via fMRI.

Contrary to our hypothesis, we observed no between-group differences in changes in appetite-related hormones over the 12-month intervention. However, consistent with physiological adaptations to weight loss, fasting leptin decreased, and ghrelin increased in both groups, reflecting known endocrine responses to negative energy balance [4]. These results align with findings from an 8-week ADF trial, which reported significant reductions in fasting leptin and increases in fasting ghrelin, yet no compensatory rise in hunger and increased sensations of fullness and PYY [58]. This suggests that IMF may enhance satiety despite endocrine changes that typically promote compensatory eating. Although appetite hormones are highly dynamic and sensitive to short-term meal timing, our fasted-state assessments at baseline and months 6 and 12 likely captured chronic diet-induced adaptations. Adiponectin and the adiponectin/leptin ratio also showed significant time effects, suggesting modest improvements in metabolic homeostasis. The adiponectin/leptin ratio, considered a sensitive indicator of insulin sensitivity and cardiometabolic risk [31], improved in both groups, possibly reflecting enhanced leptin sensitivity. Our results are consistent with a recent meta-analysis showing that IMF increases adiponectin while lowering leptin and ghrelin [29,31]. BDNF levels declined transiently at month 6 in the 4:3 IMF group, but there were no significant main effects over 12 months. Prior reviews have reported mixed findings regarding the impact of IMF and caloric restriction on BDNF and cognitive outcomes in humans [34]. Taken together, these findings suggest that while modest hormonal adaptations occurred, the superior outcomes in weight loss and eating behaviors in 4:3 IMF may be more strongly attributed to behavioral and psychological mechanisms than to differences in appetite-regulating hormones.

The results of the study should be interpreted in the context of several limitations. First, this secondary analysis was not specifically powered to detect differences in eating behaviors or appetite-related hormones, which may limit the interpretation of some findings. Some null findings may reflect limited statistical power rather than true absence of effect. Results from these analyses should be interpreted cautiously and viewed as hypothesis-generating rather than confirmatory because of the lack of adjustment for multiple comparisons across the five eating behavior and appetite-related hormone outcomes, increasing the risk of type I error. The appetite-related hormones were only measured in the AM fasted state at baseline and months 6 and 12. The absence of a standardized meal challenge and postprandial measurements of appetite hormones limits the ability to draw conclusions about the dynamic regulation of appetite hormones in response to food intake, which is critical for understanding energy balance regulation in free-living conditions. Participants were predominantly middle aged, female, and non-Hispanic White, free of eating disorders and significant cardiometabolic disease, which limits the generalizability of findings to younger and older individuals, individuals of other racial and ethnic groups, and individuals with eating disorders or other comorbid conditions. In addition, the study was conducted in the United States, further limiting applicability to international, religious, or culturally distinct populations. Despite these limitations, the study possesses several key strengths. It is the first long-term study to evaluate the effects of 4:3 IMF and DCR on a comprehensive range of eating behaviors and appetite hormones. The large sample size, randomized design, and 12-month timeframe enhance the internal validity and generalizability of the findings. Additionally, the use of validated, standardized questionnaires enabled consistent and reliable measurement of eating behaviors across participants and time points, facilitating meaningful comparisons between groups. Moreover, the incorporation of structured group counseling across both intervention arms added a critical behavioral support element that likely contributed to the high retention rate and successful weight loss.

## 5. Conclusions

In summary, the findings of this secondary analysis suggest that 4:3 IMF produces greater improvements in maladaptive eating behaviors compared to DCR over 12 months, specifically, reductions in binge eating behaviors and uncontrolled eating. Behavioral improvements were associated with greater weight loss in the 4:3 IMF group, suggesting that psycho-behavioral factors may contribute to diet adherence and weight loss outcomes. While no between-group differences were observed in fasting appetite-related hormones, our study was not designed to assess dynamic, postprandial hormone responses, which may still play a role in mediating outcomes. Notably, this is one of the first studies to evaluate the changes in eating behaviors and appetite hormones over 12 months of an IMF intervention, underscoring the importance of long-term behavioral adaptations. Future studies incorporating more frequent hormone assessment and standardized meal challenges are warranted to clarify the relative contributions of endocrine and behavioral pathways in IMF regimens. Future research should include assessment of appetite hormone response to meals in 4:3 IMF vs. DCR, as well as examine the neurocognitive and reward-based mechanisms underlying greater dietary adherence to 4:3 IMF.

## Figures and Tables

**Figure 1 nutrients-17-02385-f001:**
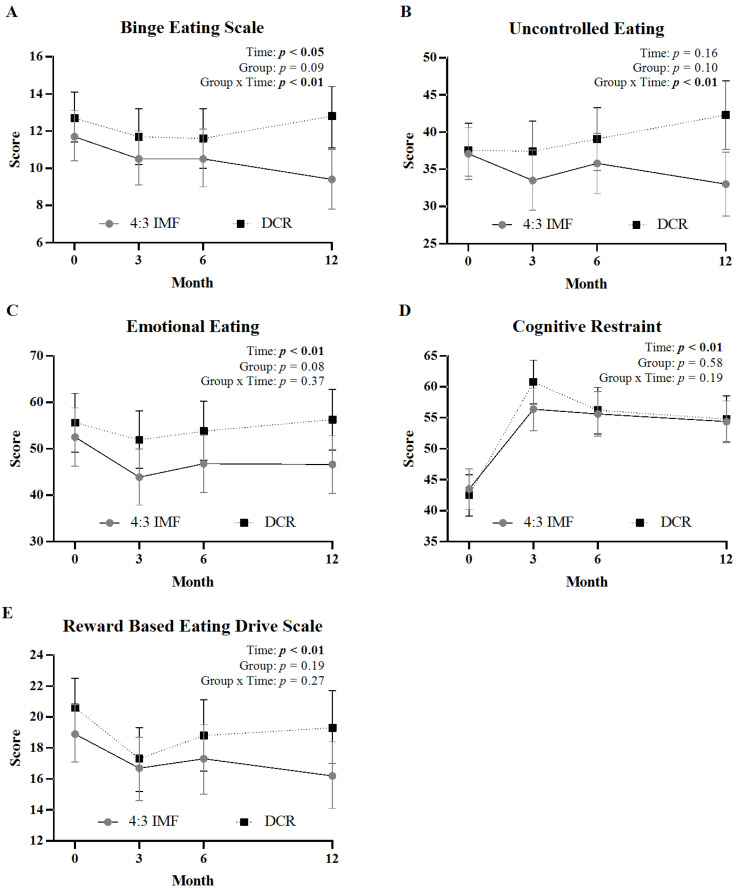
Changes in eating behavior questionnaire scores by intervention group over 12 months. Panels show mean (±95% CI) at months 0, 3, 6, and 12 for binge eating (**A**), uncontrolled eating (**B**), emotional eating (**C**), cognitive restraint (**D**), and reward-based eating (**E**) between 4:3 IMF and DCR. Statistical effects of time, group, and group-by-time interaction are displayed in the top right corner of each panel; significant effects (*p* < 0.05) are shown in bold.

**Figure 2 nutrients-17-02385-f002:**
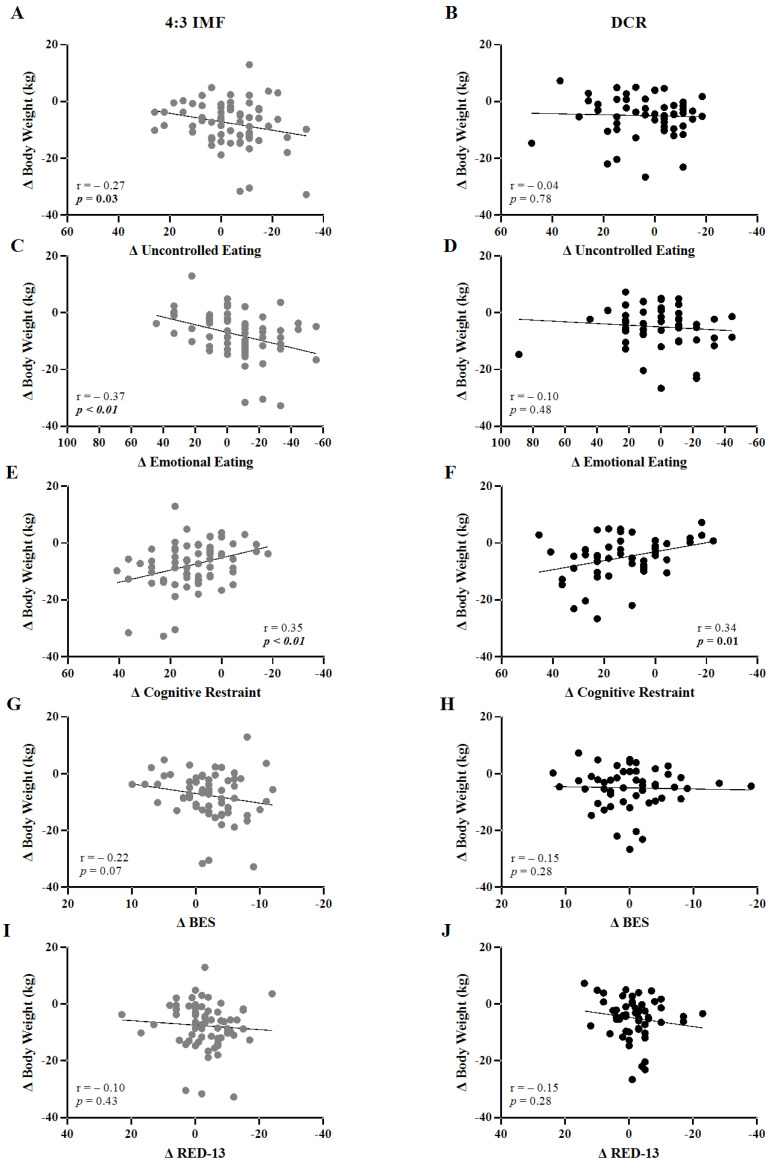
Correlations between changes in body weight, TFEQ-R18, BES, and RED-13 scores over 12 months by intervention group. Panels show scatter plots of individual participants with change in body weight (month 12–month 0) plotted against change in eating behavior scores (month 12–month 0). Data are shown for 4:3 IMF (uncontrolled eating, (**A**); emotional eating, (**C**); cognitive restraint, (**E**), BES, (**G**); and RED-13, (**I**)) and DCR (uncontrolled eating, (**B**); emotional eating, (**D**); cognitive restraint, (**F**); BES, (**H**); and RED-13, (**J**)). Pearson correlation coefficients (r) and *p*-values are shown in the lower corners of each panel, with significant *p*-values (*p* < 0.05) in bold. Abbreviations are as follows: 4:3 Intermittent Fasting (4:3 IMF), Daily Caloric Restriction (DCR), Binge Eating Scale (BES), Reward-based Eating Drive Scale, Revised 13-item (RED-13).

**Table 1 nutrients-17-02385-t001:** Baseline Characteristics.

Characteristic	4:3 IMF (n = 84)	DCR (n = 81)
Age, years	42 (10)	42 (8)
Sex		
Female	73.8%	74.1%
Male	26.2%	25.9%
Race (%)		
White	85.7%	86.4%
Asian	7.1%	2.5%
Black	6.0%	6.2%
Other	1.2%	4.9%
Ethnicity (%)		
Hispanic or Latino	17.9%	29.6%
Not Hispanic or Latino	82.1%	70.4%
Body Composition		
BMI, kg/m^2^	34.3 (4.4)	33.9 (4.4)
Weight, kg	99.2 (16.0)	95.5 (16.0)
Total Body Fat Mass (%)	40.6 (7.8)	41.1 (7.2)
Eating Behaviors (95% CI)		
BES Score	11.7 (10.4 to 13.1)	12.7 (11.4 to 14.1)
TFEQ-R18 Subscale Scores		
Uncontrolled Eating	37.1 (33.6 to 40.6)	37.6 (34.1 to 41.2)
Emotional Eating	52.5 (46.3 to 58.8)	55.6 (49.2 to 61.9)
Cognitive Restraint	43.5 (40.2 to 46.8)	42.5 (39.1 to 45.8)
RED-13 Score	18.9 (17.1 to 20.8)	20.6 (18.8 to 22.5)
Appetite Hormones (95% CI)		
Leptin (ng/mL)	68.3 (59.7 to 76.9)	67.7 (58.9 to 76.5)
Ghrelin (pg/mL)	788.0 (721.5 to 854.4)	795.5 (727.8 to 863.2)
PYY (pg/mL)	97.2 (89.5 to 104.9)	95.8 (87.9 to 103.6)
BDNF (ng/mL)	17.4 (15.7 to 19.1)	16.0 (14.3 to 17.8)
Adiponectin (µg/mL)	8.2 (7.1 to 9.2)	9.8 (8.7 to 10.9)
Adiponectin (µg/mL)/Leptin (ng/mL) Ratio	0.18 (0.14 to 0.22)	0.23 (0.18 to 0.27)

Results are mean (SD), unless otherwise indicated. Abbreviations are as follows: 4:3 Intermittent Fasting (4:3 IMF); Daily Caloric Restriction (DCR); Body Mass Index (BMI); kilogram (kg); Binge Eating Scale (BES); Three-Factor Eating Questionnaire—Revised 18-item (TFEQ-R18); Reward-based Eating Drive Scale, Revised 13-item (RED-13); peptide YY (PYY); brain-derived neurotrophic factor (BDNF).

**Table 2 nutrients-17-02385-t002:** Number of participants (n) who completed eating behavior questionnaires and provided blood samples at each timepoint, by intervention group.

Characteristic	4:3 IMF				DCR			
	M0	M3	M6	M12	M0	M3	M6	M12
Eating Behaviors								
BES	84	78	72	68	81	74	65	53
TFEQ-R18	83	76	71	68	81	74	65	52
RED-13	84	78	72	68	81	73	65	53
Appetite Hormones								
Leptin	83		66	65	79		56	52
Ghrelin	83		66	65	80		57	52
PYY	83		66	65	80		57	52
BDNF	82		66	65	78		57	52
Adiponectin	78		66	65	70		57	52

Abbreviations are as follows: 4:3 Intermittent Fasting (4:3 IMF); Daily Caloric Restriction (DCR); Months 0, 3, 6, and 12 (M0, M3, M6, and M12); Binge Eating Scale (BES); Three-Factor Eating Questionnaire—Revised 18-item (TFEQ-R18); Reward-based Eating Drive Scale, Revised 13-item (RED-13); peptide YY (PYY); brain-derived neurotrophic factor (BDNF).

**Table 3 nutrients-17-02385-t003:** Change in appetite-related hormones by randomized groups over 12 months.

Variable	Month	4:3 IMF	Change from Baseline in 4:3 IMF	DCR	Change from Baseline in DCR	Difference in Change from Baseline Between Groups	Main Effects and Interaction
Leptin (ng/mL)	0	68.3 (59.7 to 76.9)		67.7 (58.9 to 76.5)			T: <0.01
	6	43.4 (36.4 to 50.5)	−24.9 (−31.9 to −17.8)	45.6 (38.2 to 53.0)	−22.1 (−29.5 to −14.7)	2.8 (−7.5 to 13.0)	G: 0.71
	12	47.5 (40.1 to 54.9)	−20.8 (−27.5 to −14.1)	51.5 (43.6 to 59.3)	−16.2 (−23.4 to −9.6)	4.6 (−5.2 to 14.4)	GxT: 0.63
Ghrelin (pg/mL)	0	788.0 (721.5 to 854.4)		795.5 (727.8 to 863.2)			T: <0.01
	6	902.2 (823.1 to 981.3)	114.2 (59.2 to 169.3)	872.4 (790.0 to 954.7)	76.8 (18.2 to 135.4)	−37.4 (−117.8 to 43.0)	G: 0.71
	12	921.7 (839.0 to 1004.4)	133.7 (81.6 to 185.8)	886.6 (799.3 to 973.9)	91.1 (33.3 to 148.9)	−42.7 (−120.5 to 35.2)	GxT: 0.53
PYY (pg/mL)	0	97.2 (89.5 to 104.9)		95.8 (87.9 to 103.6)			T: 0.31
	6	92.9 (86.5 to 99.3)	−4.3 (−12.0 to 3.3)	92.1 (85.4 to 98.9)	−3.6 (−11.6 to 4.4)	0.7 (−10.4 to 11.8)	G: 0.70
	12	97.1 (89.7 to 104.6)	−0.1 (−8.3 to 8.2)	94.5 (86.2 to 102.8)	−1.3 (−10.3 to 7.8)	−1.2 (−13.5 to 11.0)	GxT: 0.95
BDNF (ng/mL)	0	17.4 (15.7 to 19.1)		16.0 (14.3 to 17.8)			T: 0.14
	6	14.7 (12.9 to 16.6)	−2.7 (−5.0 to 0.4)	15.5 (13.5 to 17.4)	−0.5 (−3.0 to 1.9)	2.1 (−1.2 to 5.5)	G: 0.78
	12	15.4 (13.6 to 17.2)	−2.0 (−4.4 to 0.4)	15.4 (13.4 to 17.4)	−0.6 (−3.2 to 2.0)	1.4 (−2.1 to 4.9)	GxT: 0.44
Adiponectin (μg/mL)	0	8.2 (7.1 to 9.2)		9.8 (8.7 to 10.9)			T: <0.01
	6	8.6 (7.5 to 9.6)	0.4 (−0.2 to 1.0)	10.3 (9.2 to 11.4)	0.6 (−0.1 to 1.2)	0.2 (−0.7 to 1.1)	G: <0.05
	12	10.2 (8.9 to 11.4)	2.0 (1.3 to 2.7)	11.6 (10.3 to 12.9)	1.8 (1.0 to 2.6)	−0.2 (−1.3 to 0.9)	GxT: 0.74
Adiponectin/ Leptin Ratio	0	0.18 (0.14 to 0.22)		0.23 (0.18 to 0.27)			T: <0.01
	6	0.49 (0.31 to 0.67)	0.31 (0.15 to 0.48)	0.45 (0.26 to 0.64)	0.22 (0.05 to 0.40)	−0.09 (−0.33 to 0.15	G: 0.80
	12	0.49 (0.34 to 0.64)	0.31 (0.18 to 0.44)	0.42 (0.27 to 0.58)	0.20 (0.05 to 0.34)	−0.11 (−0.31 to 0.08)	GxT: 0.54

Results are means (95% CIs) from linear mixed effects model with unstructured covariance using an intent-to-treat analysis; Statistically significant (*p* < 0.05) changes from baseline are indicated in bold. Abbreviations are as follows: Peptide YY (PYY), Brain-derived Neurotrophic Factor (BDNF), 4:3 Intermittent Fasting (4:3 IMF), Daily Caloric Restriction (DCR), time effects (T), group effects (G), groups (4:3 IMF and DCR) by time (month 0 and month 12) interaction (GxT).

## Data Availability

The original contributions presented in this study are included in the article. Further inquiries can be directed to the corresponding author.

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
