# Peer review of "Effects of 4:3 Intermittent Fasting on Eating Behaviors and Appetite Hormones: A Secondary Analysis of a 12-Month Behavioral Weight Loss Intervention"

_nutrients, 2025, doi:10.3390/nu17142385_

Round 1
Reviewer 1 Report
Comments and Suggestions for Authors
I believe the work conducted by Breit and co-workers can be considered after the following revisions are done:
Firstly, the authors have to work to rewrite their manuscript to decrease the similarities with other published papers. The current similarity index is too high.
Abstract: The first subsection should be “Background/Objectives” and it should reflect exactly this. In its current state, it is not appropriate. First, the authors need to provide a background statement and the justification to conduct the research and then to mention the aims of their work. The number of participants should be placed in the Results and not in the Methods. At the end, future perspectives should be added.
The references are not formatted according to the journal’s guidelines.
The Introduction has to be expanded and improved. More data have to be included, as well as a stronger background to support the realization of the research. The study’s relevance and novelty need to be clarified from a worldwide perspective.
The number of study participants should be moved to the Results section.
How did you consider your sample size representative of the study population? Did you perform any power analysis or other statistical analysis to verify this?
Table 1 should be formatted like Table 2 and not as an image.
The Results and Discussion are fine. However, the study’s limitations should be better discussed, preferably in a separated section/subsection
Elaborate on your Conclusions by adding some directions for further investigations.
Author Response
Reviewer 1 Comments:
I believe the work conducted by Breit and co-workers can be considered after the following revisions are done:
- Firstly, the authors have to work to rewrite their manuscript to decrease the similarities with other published papers. The current similarity index is too high.
We have received the iThenticate report from the editor and thank the editor for clarity around the similarity index. We received confirmation that the duplication rate (30% for the entire paper and <5% for a single paper) of the original manuscript is acceptable. We reduced overlapping text in the Materials and Methods section (Page 3) to reduce the duplication rate further.
- Abstract: The first subsection should be “Background/Objectives” and it should reflect exactly this. In its current state, it is not appropriate. First, the authors need to provide a background statement and the justification to conduct the research and then to mention the aims of their work. The number of participants should be placed in the Results and not in the Methods. At the end, future perspectives should be added.
Thank you for your helpful feedback. We revised the Abstract to include a clear background and justification for the study, followed by the Specific Aims. The number of participants has been moved to the Results section, and a brief future perspective has been added to the Conclusion.
- The references are not formatted according to the journal’s guidelines.
Thank you for noting this. The in-text citations and reference list have been reformatted to align with the Nutrients’ guidelines and in accordance with the MDPI ACS style.
- The Introduction has to be expanded and improved. More data have to be included, as well as a stronger background to support the realization of the research. The study’s relevance and novelty need to be clarified from a worldwide perspective
We appreciate this suggestion. The Introduction has been expanded (Page 2) to provide a more worldwide perspective. The study’s relevance and novelty have been clarified. We also clarified the study’s relevance and novelty from a global perspective, highlighting the growing interest in IMF as a behavioral weight loss strategy.
- The number of study participants should be moved to the Results section.
Thank you for this helpful suggestion. We have moved the information regarding the total number of randomized participants to the beginning of the Results section (Section 3.1, Page 5). We also reference the CONSORT diagram provided in the parent trial publication (11) to detail participant flow and attrition.
- How did you consider your sample size representative of the study population? Did you perform any power analysis or other statistical analysis to verify this?
We thank the reviewer for this important question. This secondary analysis was exploratory in nature and did not include a formal power calculation for the eating behavior or appetite hormone outcomes. However, we utilized all available blood sample and questionnaire data from participants enrolled in the parent randomized controlled trial (DRIFT). The parent trial was powered at 90% (α = 0.05) to detect a between-group difference of 3 kg in body weight at 12 months. Although this secondary analysis was not powered for the stated outcomes, the inclusion of the full dataset from the original cohort helps ensure that the analytic sample reflects the characteristics of the broader study population. We did not conduct additional post-hoc power analysis for the secondary outcomes analyzed in this manuscript because it is of little value for interpreting the results as discussed by Goodman [1] and Hoening [2]. Caveats of potential lack of power have been added to the limitations section.
- Table 1 should be formatted like Table 2 and not as an image
Thank you for this suggestion. We have uploaded an editable version of all tables for the re-submission. Table 1 presents baseline data, which are best displayed in a traditional format for clarity and group comparison. Table 2 presents longitudinal data (sample size by group and time point), which requires a different structure and accounts for the differing formats. We have edited headings in Table 2 to appear similar to Table 1.
- The Results and Discussion are fine. However, the study’s limitations should be better discussed, preferably in a separated section/subsection
We appreciate the reviewer’s suggestion regarding the discussion of study limitations. Given that the current manuscript does not include subsection headings within the Discussion, we elected not to create a separate limitations subsection this sole purpose and to maintain structural consistency and flow. However, we have expanded and clarified the limitations within the existing Discussion (Page 14) to address this important aspect more comprehensively.
- Elaborate on your Conclusions by adding some directions for further investigations.
We thank the reviewer for this suggestion. In response, we have expanded the Conclusion section (Page 14) to include directions for future research.
Reviewer 2 Report
Comments and Suggestions for Authors
The original article, "Effects of 4:3 Intermittent Fasting on Eating Behaviors and Appetite Hormones: A Secondary Analysis of a 12-Month Behavioral Weight Loss Intervention," evaluated the effect of intermittent fasting as compared to caloric restriction on eating behavior using three questionnaires (Three-Factor Eating Questionnaire (TFEQ-R18), Binge Eating Scale (BES), and Reward-based Eating Drive (RED-13) at baseline and months 3, 6, and 12). Furthermore, this effect was related to changes in appetite hormones.
In the introduction section, the authors presented the limitations of the current diets focused on caloric restriction and the known benefits and limitations of different types of intermittent fasting on appetite hormones. The study aims to expand current knowledge of the behavioral and hormonal control of food behavior. The 4:3 IF is not a very usual type of fasting. I believe that providing an extended explanation of the reasons for choosing this type of intervention would enhance the overall quality of the article.
The Methods Section is presented, describing the participants, the criteria used for inclusion and exclusion from the intervention, the questionnaires applied, and the hormones determined. However, this section would benefit from presenting the sample size calculation and the withdrawal rate from each type of intervention.
The results have shown a time effect of both CR and IF on leptin, ghrelin, PYY, and BDNF, with no differences between the groups. However, the adiponectin level has shown a significant change in the IF group. The images of the tables and figures, and their legends, are blurred and should be improved. Please include in Table 1 the significance level and the tests for significant differences between groups.
The discussion section is well-constructed, adding additional explanations of the results of the study.
Author Response
Reviewer 2 Comments:
The original article, "Effects of 4:3 Intermittent Fasting on Eating Behaviors and Appetite Hormones: A Secondary Analysis of a 12-Month Behavioral Weight Loss Intervention," evaluated the effect of intermittent fasting as compared to caloric restriction on eating behavior using three questionnaires (Three-Factor Eating Questionnaire (TFEQ-R18), Binge Eating Scale (BES), and Reward-based Eating Drive (RED-13) at baseline and months 3, 6, and 12). Furthermore, this effect was related to changes in appetite hormones.
- In the introduction section, the authors presented the limitations of the current diets focused on caloric restriction and the known benefits and limitations of different types of intermittent fasting on appetite hormones. The study aims to expand current knowledge of the behavioral and hormonal control of food behavior. The 4:3 IF is not a very usual type of fasting. I believe that providing an extended explanation of the reasons for choosing this type of intervention would enhance the overall quality of the article.
We appreciate this thoughtful suggestion. In response, we have added a rationale to the Introduction (Page 2) discussing the selection of the 4:3 IMF regimen. Specifically, we note that 4:3 IMF offers an intermediate approach between 5:2 IMF and ADF, two of the most popular forms of IMF, potentially enhancing behavioral feasibility. We also added references (6-10) to support this context and highlight the novelty of evaluating the effects of 4:3 IMF on eating behaviors and appetite.
- The Methods Section is presented, describing the participants, the criteria used for inclusion and exclusion from the intervention, the questionnaires applied, and the hormones determined. However, this section would benefit from presenting the sample size calculation and the withdrawal rate from each type of intervention.
Please see our response to Reviewer #1,above. As this is a secondary analysis, an a-priori formal power calculation or post-hoc power calculation was not conducted.
- The results have shown a time effect of both CR and IF on leptin, ghrelin, PYY, and BDNF, with no differences between the groups. However, the adiponectin level has shown a significant change in the IF group. The images of the tables and figures, and their legends, are blurred and should be improved. Please include in Table 1 the significance level and the tests for significant differences between groups.
We appreciate this comment, however for consistency with the format of the previously published results of the parent randomized trial [3], p-values are not included in Table 1 for comparisons between groups. The purpose of Table 1 is to describe the baseline characteristics of the study population, not to engage in inferential analysis and hypothesis testing. Randomization distributes known and unknown confounding factors, and any differences seen will be due to chance. Showing p-values can give a misleading picture of these differences [4]. Additionally, we have analyzed the data using linear mixed effect models under which contrasts were set up to assess the difference in change score in order to assess intervention effect. This approach has taken the correction between repeated measures into account and appropriately adjusted for baseline imbalance. We would be happy to include p-values if Nutrients mandates p-values in Table 1.
- The discussion section is well-constructed, adding additional explanations of the results of the study.
We thank the reviewer for this comment.